# Relationship between Oral Parafunctional and Postural Habits and the Symptoms of Temporomandibular Disorders: A Survey-Based Cross-Sectional Cohort Study Using Propensity Score Matching Analysis

**DOI:** 10.3390/jcm11216396

**Published:** 2022-10-28

**Authors:** Susumu Abe, Fumiaki Kawano, Yoshizo Matsuka, Taeko Masuda, Toshinori Okawa, Eiji Tanaka

**Affiliations:** 1Department of Comprehensive Dentistry, Graduate School of Biomedical Sciences, Tokushima University, Tokushima 770-8504, Japan; 2Department of Stomatognathic Function and Occlusal Reconstruction, Graduate School of Biomedical Sciences, Tokushima University, Tokushima 770-8504, Japan; 3Department of Orthodontics and Dentofacial Orthopedics, Graduate School of Biomedical Sciences, Tokushima University, Tokushima 770-8504, Japan

**Keywords:** temporomandibular disorders, oral parafunctional habit, cross sectional cohort study, propensity score matching analysis, postural habit

## Abstract

Temporomandibular disorders (TMDs) are a multifactorial joint disease of the masticatory system. The possible etiological factors involved in the onset and progression of TMDs include oral parafunctional habits (OPFHs) and postural habits (PHs). However, little information is available on the association between OPFHs and PHs and a predisposition to TMDs. Thus, to investigate whether the presence of OPFH and PH predisposes individuals to TMDs, a survey-based cross-sectional cohort study of self-reported TMD was performed. A total of 2292 patients with TMD were recruited for the survey. Through one-to-one propensity score matching, 166 patients with and without sleep bruxism (SB) were selected. The SB group had a significantly higher risk of masticatory muscular pain or fatigue than the non-SB group (*p* = 0.018). Furthermore, the SB group without other OFPHs and PHs did not show a significantly higher risk of TMD symptoms than the non-SB group. Diurnal clenching and bad posture also affect the stomatognathic system, causing pain or fatigue; however, it did not result in TMDs in patients without any OPFHs and PHs. This implies that OPFHs and PHs may increase the risk of TMD symptoms in coexistence with other habits.

## 1. Introduction

Temporomandibular disorders (TMDs) are a collective term used to describe pain and functional disturbance of the masticatory system [1,2]. Masticatory system pain and disturbance affect oral health-related quality of life [3]. Furthermore, prolonged pain may affect the overall quality of life of the patient [4]. Thus, clinicians should aim for rapid elimination of TMD-related pain to allow for improvement in masticatory function.

According to the Diagnostic Criteria for TMD (DC/TMD) Axis I, TMDs can be divided into pain-related TMDs including masticatory muscle disorders and arthralgia, and intra-articular disorder including disc displacement with or without reduction and mouth opening limitation and osteoarthritis [5]. Epidemiological surveys have shown that TMD is the second most common musculoskeletal disorder that causes pain and disability [6,7] and that musculoskeletal pain occurs in 2–12% of the population [8,9,10], while 10% of the population reported severe symptoms [11]. In a large population study, tenderness of the TMJ and masticatory muscle was observed in 5% and 15% of the population, respectively [12]; however, these symptoms were self-reported by approximately 4% and 6–8% of the patients, respectively [13,14]. The reported prevalence of TMD symptoms and signs may differ widely depending on the method of examination, diagnostic criteria, age, and patient sample selection [15,16,17]. The commonly used treatments for muscular and arthrogenous TMDs are clarified into four types: non-invasive management modalities including occlusal splint, medications, orthotics, and physical therapy; minimally invasive modalities including arthrocentesis, arthroscopy, and intra-articular injections of corticosteroids or hyaluronic acid; invasive surgical modalities including arthroplasty, osteotomy, and osteodistraction; and salvage procedures including total joint replacement [18,19].

Numerous factors can be considered in the etiology of TMDs. Oral parafunctional habits (OPFHs), such as bruxism, diurnal clenching, and thumb sucking, are common and usually have negligible effects on the stomatognathic apparatus [20]. Sleep bruxism (SB) is a movement disorder characterized by repetitive masticatory muscle activity during sleep, and it could be associated with myofascial pain and arthralgia [21]. However, when the activity exceeds the individual resistance threshold or the host’s adaptive capacity decreases, it may cause damage to the dentition, muscles, and TMJ [22]. Several researchers have suggested that OPFHs are a possible etiological factor in the progression of TMDs in growing populations [22,23,24,25]. Furthermore, postural habits (PHs), such as bad posture and sleeping in face down position, are considered indicative of a potential TMD [26,27]. However, little information is available on the association between OPFHs and PHs and a predisposition to TMDs.

Thus, to investigate whether the presence of OPFH and PH predisposes to TMDs, we performed a survey-based cross-sectional cohort study of self-reported TMD among outpatients who visited the Temporomandibular Joint Clinic of Tokushima University Hospital. We aimed to assess the prevalence of self-reported subjective symptoms indicative of TMD and their correlation with OPFHs and PHs associated with the onset and progression of TMDs.

## 2. Materials and Methods

### 2.1. Participants

Outpatients with the chief complaint of TMD symptoms who visited the Temporomandibular Joint Clinic at Tokushima University Hospital between April 2010 and September 2021 were recruited. The inclusion criteria for the patients were age >12 years and at least one TMD symptom according to the DC/TMD [5]. The exclusion criteria were uncertain symptoms of TMD; orofacial pain, such as trigeminal neuralgia; toothache; and pain of dental origin.

The Ethics Committee of Tokushima University Hospital approved the study, and written informed consent was obtained from each patient after sufficient explanation of the research purposes (No. 2279-4).

### 2.2. Questionnaire

All participants were asked to fill out a unique self-reported questionnaire, including items on TMD symptoms, general and oral health, OPFHs and PHs, and some sociodemographic issues. The answer alternatives to all questions were “yes” or “no”, except for items 9 and 14 (Table 1). All patients answered the prepared 21 items in the self-reported questionnaire before the clinical examination. Of the 21 items, 6 items evaluated the present TMD symptoms for the patients, and the remaining included the past medical history or present/past OPFH and PH. In addition, multiple-choice questions regarding the following topics were added to the questionnaire: TMD symptoms (TMJ pain during various mandibular movements (item 1); masticatory muscle pain and fatigue (item 2); difficulty in mouth opening (item 3); TMJ sounds, such as clicking or popping (item 4); frequent headaches and stiff shoulder (item 5); and joint luxation (item 6)), and OPFH and PH (SB (item 16), diurnal grinding or clenching (DC) (item 17), chin on hand (CH) (item 18), face down (FD) (item 19), and bad posture (BP) (item 20)). Furthermore, two items in the questionnaire about previous TMJ treatment (item 9) and chewing side (item 13) with three answer choices were included. Patients were asked to grade two items on “maximum present pain related to this symptom (item 7)” and “degree of daily hindrance related to this symptom (item 8)”. These were rated on a visual analog scale (VAS) with a 100 mm straight horizontal line anchored by descriptors at each end. For “maximum present pain related to this symptom”, which measured painful experiences, the descriptor was “no pain” on the left end and “worst pain” on the right end; for the “degree of daily hindrance related to this symptom”, it was “not at all” on the left and “hardest time” on the right end.

### 2.3. Setting of Outcomes

The clinical outcomes were set as the present TMD symptoms, which included “TMJ pain during various mandibular movements (item 1)”, “masticatory muscle pain and fatigue (item 2)”, “difficulty in mouth opening (item 3)”, “TMJ sounds, such as clicking or popping (item 4)”, “frequent headaches and stiff shoulder (item 5)”, and “joint luxation (item 6)” from answers of alternatively self-reported questionnaires.

### 2.4. Statistical Analysis

In the present study, the estimated propensity score of each patient was used to perform individual matching analysis to determine whether OPFH and PH were present. Statistical analysis was performed using SPSS 27.0 (SPSS Inc., Chicago, IL, USA). To calculate the propensity score, a logistic regression model was fitted for each symptom and the patients’ constitution, including age, sex, habits, and history. To evaluate the effect of OPFH and PH on TMD symptoms, the estimated propensity score is required to calculate C-statistics, which is represented by the area under the receiver operating characteristic (ROC) curve and is an indicator for evaluating the goodness of fit [28]. Each patient who had an OPFH or PH was matched with a patient who had no habits with the closest estimated propensity on the logit scale within a specified range (≤0.2 of the pooled standard deviation of estimated logits) as a caliper using a specific OPFH or PH. The propensity score was calculated in the absence of other habits except for the targeted habit.

Fisher’s exact test was used to compare the six present TMD symptoms between the groups with and without OPFHs and PHs. Logistic regression analysis for each TMD symptom was performed in propensity-matched patients to analyze the odds ratios and 95% confidence intervals with reference to the absence of OPFHs and PHs. The two VAS scales for pain and daily hindrance were compared between the presence and absence of each habit using the Mann–Whitney U test. 

In the present study, two different methods were used to analyze the effect of OPFHs and PHs: (1) all items, including other OPFHs and PHs except for targeted habits were used, and (2) patients without other OPFHs and PHs were selected to investigate the effect of targeted habits on the onset and progression of TMD symptoms. 

## 3. Results

Of 2367 outpatients, 2292 patients with TMD (639 males and 1653 female) were recruited for the survey using the self-reported questionnaire (Figure 1). The remaining 75 patients who did not fill in the questionnaire items, responded to an indefinite answer, or were below 12 years of age were excluded. Because this analysis focused on each OPFH or PH, one-to-one propensity score matching was performed for each habit to patient history. 

Table 2 shows the demographic and clinical characteristics of TMD patients with and without SB. Nine items, including age, medical history of TMD, orthodontic treatment experience, nervousness, hobby, and involvement in sports, and four OPFHs were significantly different between TMD groups with and without SB before propensity score matching (Table 2-(1)). As these items were regarded as confounding factors, we performed propensity score matching to remove these factors. A total of 904 patients with and without SB were selected, and the C-statistic for goodness of fit was calculated using the area under the ROC curve. The computed value was 0.683 in the propensity score model. After propensity score matching, patient distributions were closely balanced (Table 2-(2)). It was observed that the values for all items did not differ significantly between the TMD patients with and without SB. 

To examine whether SB affects TMD symptoms, patients without other OPFHs and PHs were included (Table 3). A total of 675 patients had no OPFHs or PHs regardless of SB presence. Significant differences in age and habitual chewing side were found between the patients with and without SB (Table 3). Through one-to-one propensity score matching, 166 patients with and without SB were selected. The C-statistic for goodness of fit in the propensity score model was calculated as 0.620.

Table 4 shows the results of the logistic regression analysis of TMD symptoms involving SB presence in the propensity-matched group in comparison with the non-SB group. The SB group had a significantly higher risk of masticatory muscular pain or fatigue compared with the non-SB group (odds ratio = 1.523; 95% confidence interval, 1.074–2.159; *p* = 0.018), including medical history, OPFHs, and PHs. However, the degree of maximum pain of daily hindrance did not differ significantly between the SB and non-SB groups, regardless of masticatory muscle pain or fatigue. Furthermore, the SB group without other OFPHs and PHs showed no significantly higher risk of TMD symptoms than the non-SB group (Table 4).

For other OFPHs or PHs except SB, the demographic and clinical characteristics of TMD patients with and without DC, CH, FD, or BP were evaluated (Appendix A). Before propensity score matching, the results indicated significant differences in items 5 to 7 between TMD patients with and without DC, CH, FD, or BP. After propensity score matching, patient distributions were closely balanced between TMD patients with and without each OPFH or PH. Furthermore, to identify the single habit as the causative factor for TMD, single-targeted habits, except the other OFPHs and PHs, were analyzed (Appendix A). For each single targeted habit (DC, CH, FD, or BP), the demographic and clinical characteristics of TMD patients without other OFPHs or PHs, except for each single targeted habit, before and after propensity score matching. Before propensity score matching, items 3 to 7 showed significant differences between the presence and absence of each single targeted habit. However, there were no significant differences between the presence and absence of each single targeted habit after propensity score matching. All C-statistics for goodness of fit were calculated around 0.700 in the propensity score model. Notably, the DC group had a significantly higher risk of jaw pain (odds ratio = 1.372; 95% confidence interval, 1.063–1.770; *p* = 0.015), masticatory pain or fatigue (odds ratio = 1.477; 95% confidence interval, 1.100–1.982; *p* = 0.009), and headache or shoulder stiffness (odds ratio = 1.509; 95% confidence interval, 1.203–1.894; *p* < 0.001) than the non-DC group (Table 4). Furthermore, the DC group had significantly higher maximum pain and daily hindrance than the non-DC group (*p* = 0.003 and *p* = 0.010, respectively). However, patients with TMD without other OPFHs and PHs, except for DC, showed no significantly higher risk of TMD symptoms (Table 4). The BP group had a significantly higher risk of headache or shoulder stiffness (odds ratio = 1.599; 95% confidence interval, 1.278–2.002; *p* < 0.001) than the non-BP group (Table 4). Furthermore, TMD patients without other OPFHs and PHs, except for BP, showed a significantly higher risk of headache or shoulder stiffness (odds ratio = 1.713; 95% confidence interval, 1.160–2.532; *p* = 0.007) than the non-BP group (Table 4). However, as CH and FD did not have significant differences in all items between TMD patients with and without CH or FD, OPFHs and PHs did not affect TMD symptoms (Table 4).

## 4. Discussion

Numerous epidemiological surveys of TMD have been designed and reported previously [5,28,29,30,31]. In these studies, the prevalence of TMD was examined in general subjects or patient populations, and varying incidence rates were reported, ranging from 20% to 50% [29]. These studies also indicated that TMDs have a multifactorial etiology and that OPFHs (e.g., nocturnal bruxism and clenching), DC (teeth contact habit), anatomical factors, trauma or general hypermobility of the joints, psychosocial issues (e.g., anxiety and depressive syndrome), and cervical posture are potential risk factors [29,32,33]. However, considering the presence of confounding factors, these findings imply that participants with OPFHs are subject to TMD symptoms and signs. Thus, to avoid confounds in the study design, a propensity score matching analysis was adopted in this study. The estimated propensity score is commonly applied in clinical or epidemiological studies [34]. It allows for dealing with various selection biases in retrospective studies, such as the pseudo-randomization of data [35]. Thus, this analysis can organize confounds by indication [36]. This can be applied to the comparison of observed prognostic factors, such as the effect, cost, or treatment duration between different clinical treatment groups. The estimated propensity score involves the extraneous factor that each patient had and is suitable to match the resembling patients using a fixed caliper, which is the nearest neighbor matching method. When using the propensity score-matched analysis in this study, demographic factors, individual medical or dental history, and the presence of jobs and hobbies were unified to remove confounding factors. To our knowledge, this is the first study in which a matched case–control analysis was conducted to identify the causes of TMD.

The present study showed that SB affected pain or fatigue of the masticatory muscles, and DC affected jaw pain, pain or fatigue of the masticatory muscles [21], and headache or shoulder stiffness in matched patients with a medical history and various habits. However, in patients without OPFHs and PHs, SB and DC were not considered potential risk factors for TMD symptoms. Thus, although SB or DC alone did not affect TMD symptoms, these habits might produce a risk for TMD symptoms when combined with other habits. Previous studies reported that various oral parafunctional factors can affect TMD symptoms [22,37,38,39,40]. Our results showed that masticatory muscle pain, fatigue, headache, or shoulder stiffness might develop combined OPHFs and PHs, and these TMD symptoms are recognized as a part of myofascial pain or fatigue [5]. Although management of myofascial pain or fatigue is not sufficient to resolve pathophysiological mechanisms, excessive extension or contraction might stimulate nociceptive neurons in the skeletal muscle. Furthermore, nociceptors on the terminals of nociceptive neurons develop pain or fatigue [41]. By continuing excessive activation in the peripheral nervous system, pain sensitivity might increase, and peripheral or central nervous plasticity may develop [42].

The following considerations and limitations of the present cohort survey should be addressed. First, as this cohort survey was performed using a propensity score-matched analysis, the results obtained were generalizable to the paired data by the range of propensity scores [34]. Therefore, the propensity score-matched method can reduce or remove the biases in demographic factors or medical history but not the effect of other biases from unobserved factors, such as information on internal medication, academic history, work history/current job, and annual income may persist [43]. Our score-matched analysis should be organized using the high propensity score-matched analysis because of the lack of unobserved confounding factors. Second, although SB was regarded as an OPFH in the self-report questionnaire, previous reports provided detailed information about the patients [44], for example, whether the patient is self-aware about SB and how frequently SB occurs in a week. However, each item for OPFH or PH in this self-report questionnaire was not included in detail, and this questionnaire was not applied in a pilot study with a smaller population. Finally, the questionnaire used in this survey did not include questions about occlusal status and masticatory performance [12]. The occlusal condition is an important factor in diagnosing TMDs; therefore, further surveys and investigations involving occlusal conditions are needed to identify the specific causative factors in patients with TMD.

## 5. Conclusions

In conclusion, we performed a propensity score-matched analysis using a large cohort of TMD patients with OFPHs and/or PHs. According to the results of our matched case–control study, SB, DC, and BP affected the stomatognathic system in terms of pain or fatigue while SB and DC did not affect TMD symptoms in patients without OPFHs and PHs. This implies that these habits might produce a risk for TMD symptoms owing to the coexistence of other habits. 

## Figures and Tables

**Figure 1 jcm-11-06396-f001:**
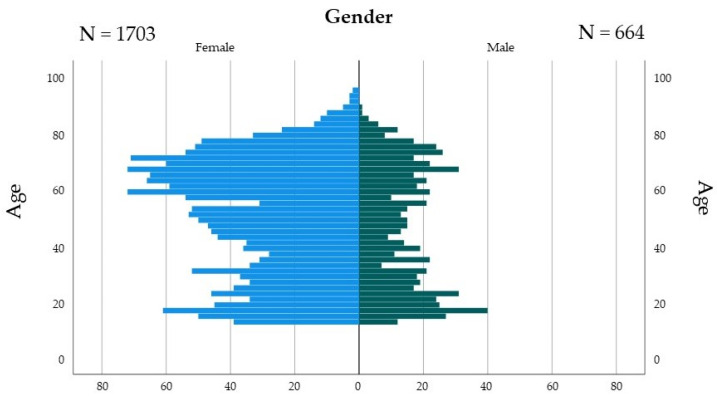
Demographic data of TMD patients for age.

**Table 1 jcm-11-06396-t001:** Questions for patients with TMD at the initial visit.

*TMJ Symptoms*
1. Do you have pain in the jaw joint?
2. Do you have masticatory muscular pain or fatigue?
3. Is it hard for you to open your mouth?
4. Do you have clicking or popping sound in either or both jaws?
5. Do you ever have frequent headaches or neck and/or shoulder pain?
6. Do you ever have difficulty closing after opening wide? (joint luxation)
7. How do you experience maximum pain related to this symptom? (Could you check the bars?)
8. How do you have a daily hindrance related to this symptom? (Could you check the bars?)
** *Medical history or characteristics* **
9. Have you ever had a previous medical history of TMD?
If you answered “Yes”, was previous medical history same or different side for presence of TMD symptom?
10. Have you ever had a previous TMD treatment?
11. Have you ever had systemic bone or joint problems, such as rheumatic arthritis, gout, osteoporosis, or osteoarthritis?
12. Have you ever injured your head, neck, jaw, or spine?
13. Have you ever had orthodontic treatment?
14. Which side do you usually or exclusively use during chewing, “Right side”, “Left side”, or “Both sides”?
15. Do you consider yourself a nervous person?
16. Do you frequently follow long-term desk work, long-time driving, hobbies, or sports?
** *Oral parafunctional or postural habits* **
17. Do you grind your teeth during your sleep? (sleep bruxism)
18. Do you clench or grind your teeth during the day? (awake bruxism or tooth contact habits)
19. Do you rest your chin on your hand?
20. Do you sleep lying face down?
21. Do you have bad posture such as a hunch?

**Table 2 jcm-11-06396-t002:** Demographic and clinical characteristics of TMD patients with and without SB. (1) Before propensity score matching. (2) After propensity score matching.

(1) Before Propensity Score Matching				
		SB		
		Yes (=Presence)	No (=Absence)	*p*–Value
Age (years)		43.6 (28–57.8) (N = 484)	53.4 (29.9–67.9) (N = 1804)	<0.001
Gender	Female	352 (21.3%)	1301 (78.7%)	0.995
(N = 2292)	Male	136 (21.3%)	503 (78.7%)	
Medical history of TMJ (N = 2211)	Yes and Same	121 (24%)	383 (76%)	<0.001
	Yes and Difference	146 (25.7%)	421 (74.3%)	
	No	205 (18%)	935 (82%)	
TMD treatment	Yes	128 (24.1%)	403 (75.9%)	0.079
(N = 2269)	No	357 (20.5%)	1381 (79.5%)	
Systemic history	Yes	95 (19%)	405 (81%)	0.153
(N = 2276)	No	390 (22%)	1386 (78%)	
Head and neck injury	Yes	89 (23.8%)	285 (76.2%)	0.205
(N = 2277)	No	397 (20.9%)	1506 (79.1%)	
Orthodontic treatment	Yes	70 (29.4%)	168 (70.6%)	0.001
(N = 2285)	No	418 (20.4%)	1629 (79.6%)	
Chewing side (N = 2257)	Both side	174 (20.1%)	691 (79.9%)	0.147
	Right side	164 (20.3%)	642 (79.7%)	
	Left side	141 (24.1%)	445 (75.9%)	
Nervous person	Yes	336 (23.3%)	1106 (76.7%)	0.003
(N = 2262)	No	147 (17.9%)	673 (82.1%)	
Desk work and recreation	Yes	294 (23.9%)	936 (76.1%)	0.002
(N = 2217)	No	183 (18.5%)	804 (81.5%)	
SB	Yes	278 (34.4%)	530 (65.6%)	<0.001
(N = 2292)	No	210 (14.2%)	1274 (85.8%)	
CH	Yes	141 (25.8%)	405 (74.2%)	0.003
(N = 2292)	No	347 (19.9%)	1399 (80.1%)	
FD	Yes	93 (25.1%)	277 (74.9%)	0.049
(N = 2292)	No	395 (20.6%)	1527 (79.4%)	
BP	Yes	243 (24.4%)	752 (75.6%)	0.001
(N = 2292)	No	245 (18.9%)	1052 (81.1%)	
**(2) After Propensity Score Matching**				
		**SB**		
		**Yes (=Presence)**	**No (=Absence)**	** *p* ** **–Value**
Age (years)		43.4 (27.4–58.4) (N = 452)	43.6 (24.1–59.8) (N = 452)	0.873
Gender	Female	330 (50.5%)	323 (49.5%)	0.603
(N = 904)	Male	122 (48.6%)	129 (51.4%)	
Medical history of TMJ (N = 904)	Yes and Same	116 (48.9%)	121 (51.1%)	0.920
	Yes and Difference	137 (50.7%)	133 (49.3%)	
	No	199 (50.1%)	198 (49.9%)	
TMD treatment	Yes	121 (48%)	131 (52%)	0.458
(N = 904)	No	331 (50.8%)	321 (49.2%)	
Systemic history	Yes	91 (50%)	91 (50%)	1.000
(N = 904)	No	361 (50%)	361 (50%)	
Head and neck injury	Yes	88 (49.7%)	89 (50.3%)	0.933
(N = 904)	No	364 (50.1%)	363 (49.9%)	
Orthodontic treatment	Yes	63 (52.1%)	58 (47.9%)	0.625
(N = 904)	No	389 (49.7%)	394 (50.3%)	
Chewing side (N = 940)	Both side	164 (47.4%)	182 (52.6%)	0.465
	Right side	150 (51.4%)	142 (48.6%)	
	Left side	138 (51.9%)	128 (48.1%)	
Nervous person	Yes	311 (49.8%)	313 (50.2%)	0.886
(N = 904)	No	141 (50.4%)	139 (49.6%)	
Desk work and recreation	Yes	278 (50.5%)	273 (49.5%)	0.733
(N = 904)	No	174 (49.3%)	179 (50.7%)	
DC	Yes	255 (49.7%)	258 (50.3%)	0.840
(N = 904)	No	197 (50.4%)	194 (49.6%)	
CH	Yes	132 (49.8%)	133 (50.2%)	0.942
(N = 904)	No	320 (50.1%)	319 (49.9%)	
FD	Yes	86 (47.8%)	94 (52.2%)	0.505
(N = 904)	No	366 (50.6%)	358 (49.4%)	
BP	Yes	221 (49.1%)	229 (50.9%)	0.595
(N = 904)	No	231 (50.9%)	223 (49.1%)	

**Table 3 jcm-11-06396-t003:** Demographic and clinical characteristics of TMD patients with and without SB, in case the other OPFHs and PHs are absent.

(1) Before Propensity Score Matching				
		SB		
		Yes (=Presence)	No (=Absence)	*p*–Value
Age (years)		54.8 (39.5–67.7) (N = 89)	63.6 (47.1–73.2) (N = 586)	0.001
Gender	Female	59 (12.7%)	404 (87.3%)	0.616
(N = 675)	Male	30 (14.2%)	182 (85.8%)	
Medical history of TMJ (N = 647)	Yes and Same	15 (12.5%)	105 (87.5%)	0.278
	Yes and Difference	23 (17.3%)	110 (82.7%)	
	No	47 (11.9%)	347 (88.1%)	
TMD treatment	Yes	21 (14.8%)	121 (85.2%)	0.527
(N = 667)	No	67 (12.8%)	458 (87.2%)	
Systemic history	Yes	22 (11.6%)	167 (88.4%)	0.479
(N = 671)	No	66 (13.7%)	416 (86.3%)	
Head and neck injury	Yes	14 (14.7%)	81 (85.3%)	0.643
(N = 672)	No	75 (13%)	502 (87%)	
Orthodontic treatment	Yes	7 (17.1%)	34 (82.9%)	0.453
(N = 673)	No	82 (13%)	550 (87%)	
Chewing side (N = 669)	Both side	33 (11.5%)	255 (88.5%)	0.021
	Right side	24 (11%)	195 (89%)	
	Left side	32 (19.8%)	130 (80.2%)	
Nervous person	Yes	53 (15.3%)	294 (84.7%)	0.131
(N = 666)	No	36 (11.3%)	283 (88.7%)	
Desk work and recreation	Yes	45 (15.6%)	243 (84.4%)	0.105
(N = 659)	No	42 (11.3%)	329 (88.7%)	
**(2) After Propensity Score Matching**				
		**SB**		
		**Yes (=Presence)**	**No (=Absence)**	** *p* ** **–Value**
Age (years)		55.9 (39.6–67.7) (N = 83)	52.9 (28.4–72.9) (N = 83)	0.626
Gender	Female	55 (50.9%)	53 (49.1%)	0.745
(N = 166)	Male	28 (48.3%)	30 (51.7%)	
Medical history of TMJ (N = 166)	Yes and Same	15 (51.7%)	14 (48.3%)	0.978
	Yes and Difference	22 (50%)	22 (50%)	
	No	46 (49.5%)	47 (50.5%)	
TMD treatment	Yes	19 (51.4%)	18 (48.6%)	0.852
(N = 166)	No	64 (49.6%)	65 (50.4%)	
Systemic history	Yes	21 (48.8%)	22 (51.2%)	0.859
(N = 166)	No	62 (50.4%)	61 (49.6%)	
Head and neck injury	Yes	14 (42.4%)	19 (57.6%)	0.331
(N = 166)	No	69 (51.9%)	64 (48.1%)	
Orthodontic treatment	Yes	6 (40%)	9 (60%)	0.417
(N = 166)	No	77 (51%)	74 (49%)	
Chewing side (N = 166)	Both side	33 (47.8%)	36 (52.2%)	0.594
	Right side	22 (46.8%)	25 (53.2%)	
	Left side	28 (56%)	22 (44%)	
Nervous person	Yes	48 (50%)	48 (50%)	1.000
(N = 166)	No	35 (50%)	35 (50%)	
Desk work and recreation	Yes	42 (51.9%)	39 (48.1%)	0.641
(N = 166)	No	41 (48.2%)	44 (51.8%)	

**Table 4 jcm-11-06396-t004:** Logistic regression analysis of TMD symptoms with the presence of targeted habit with reference to the absence of SB in the propensity-matched group. (1) Regardless of the presence or absence of other OPFHs and PHs. (2) In cases where other OPFHs and PHs are absent, except for the targeted habits.

(1)	Habit	N	Odds Ratio	95% Confidence Interval	*p*–Value
Pain in jaw joint	SB	841	1.051	0.776–1.425	0.746
	**DC**	**1228**	**1.372**	**1.063–1.770**	**0.015**
	CH	935	0.954	0.715–1.272	0.749
	FD	623	1.186	0.836–1.683	0.339
	BP	1269	0.890	0.695–1.139	0.354
Masticatory muscular pain/fatigue	**SB**	**829**	**1.523**	**1.074–2.159**	**0.018**
	**DC**	**1209**	**1.477**	**1.100–1.982**	**0.009**
	CH	908	1.127	0.782–1.623	0.521
	FD	608	0.824	0.535–1.269	0.378
	BP	1240	1.306	0.972–1.754	0.076
Hard to open mouth	SB	829	0.781	0.594–1.027	0.077
	DC	1214	1.023	0.816–1.282	0.847
	CH	913	1.045	0.804–1.357	0.742
	FD	610	1.238	0.900–1.703	0.189
	BP	1248	0.880	0.704–1.100	0.261
Clicking/popping in jaw	SB	852	0.823	0.620–1.092	0.176
	DC	1244	0.877	0.696–1.104	0.176
	CH	938	0.902	0.691–1.178	0.448
	FD	638	1.357	0.978–1.883	0.068
	BP	1291	1.169	0.930–1.469	0.180
Headache/shoulder stiffness	SB	824	1.218	0.926–1.602	0.159
	**DC**	**1206**	**1.509**	**1.203–1.894**	**<0.001**
	CH	908	0.948	0.730–1.229	0.685
	FD	606	1.169	0.850–1.610	0.337
	**BP**	**1238**	**1.599**	**1.278–2.002**	**<0.001**
Joint luxation	SB	827	0.748	0.400–1.400	0.363
	DC	1206	0.739	0.424–1.288	0.285
	CH	909	0.752	0.425–1.329	0.325
	FD	606	1.386	0.696–2.761	0.352
	BP	1237	0.870	0.520–1.457	0.596
**(1)**			**Targeted Habit**		
	**Habit**	**N**	**Yes (=Presence)**	**No (=Absence)**	** *p* ** **–Value**
Maximum Pain (VAS)	SB	888	40.00 (11.50–56.50)	40.00 (12.00–61.00)	0.548
	**DC**	**1291**	**41.50 (18.00–60.00)**	**36.00 (9.00–55.00)**	**0.003**
	CH	976	35.00 (13.00–55.00)	42.00 (13.25–60.00)	0.151
	FD	671	35.00 (12.00–56.00)	35.00 (10.00–55.00)	0.769
	BP	1346	40.00 (14.00–56.00)	40.00 (14.00–58.00)	0.616
Daily Hindrance (VAS)	SB	877	47.00 (20.00–62.00)	45.00 (20.00–60.00)	0.828
	**DC**	**1282**	**49.00 (22.00–64.00)**	**45.50 (19.00–56.00)**	**0.010**
	CH	969	40.00 (18.00–55.25)	47.00 (20.00–60.00)	0.062
	FD	670	42.00 (19.00–56.00)	40.00 (18.50–58.00)	0.887
	BP	1336	49.00 (21.25–61.75)	45.00 (20.00–57.00)	0.078
**(2)**	**Habit**	**N**	**Odds Ratio**	**95% Confidence Interval**	***p*–Value**
Pain in jaw joint	SB	149	0.844	0.375–1.901	0.683
	DC	355	1.281	0.792–2.073	0.312
	CH	136	1.040	0.479–2.260	0.921
	FD	99	1.531	0.661–3.547	0.319
	BP	430	1.017	0.672–1.540	0.935
Masticatory muscular pain/fatigue	SB	142	1.810	0.727–4.504	0.198
	DC	342	1.466	0.850–2.526	0.167
	CH	131	0.395	0.097–1.598	0.180
	FD	96	1.955	0.459–8.318	0.358
	BP	416	1.618	0.913–2.866	0.097
Hard to open mouth	SB	145	0.698	0.363–1.342	0.281
	DC	344	0.901	0.589–1.379	0.632
	CH	133	1.092	0.551–2.165	0.801
	FD	96	1.635	0.711–3.758	0.246
	BP	420	0.975	0.664–1.432	0.897
Clicking/popping in jaw	SB	152	0.820	0.429–1.568	0.549
	DC	356	0.732	0.479–1.119	0.149
	CH	140	0.530	0.270–1.041	0.064
	FD	106	0.659	0.295–1.473	0.308
	BP	444	1.052	0.714–1.551	0.798
Headache/shoulder stiffness	SB	142	1.474	0.758–2.867	0.252
	DC	342	1.060	0.692–1.624	0.790
	CH	131	0.914	0.450–1.856	0.804
	FD	96	1.497	0.650–3.446	0.342
	**BP**	**418**	**1.713**	**1.160–2.532**	**0.007**
Joint luxation	SB	143	0.250	0.027–2.294	0.187
	DC	342	0.347	0.090–1.330	0.107
	CH	131	0.774	0.198–3.022	0.712
	FD	97	0.285	0.054–1.488	0.117
	BP	416	0.532	0.218–1.297	0.159
**(2)**			**Targeted Habit**		
	**Habit**	**N**	**Yes (=Presence)**	**No (=Absence)**	** *p* ** **–Value**
Maximum Pain (VAS)	SB	162	29.50 (8.00–53.50)	34.00 (5.00–54.75)	0.916
	DC	376	42.00 (17.50–61.00)	35.00 (10.00–55.00)	0.068
	CH	144	30.00 (12.50–51.50)	35.00 (3.00–53.00)	0.997
	FD	105	35.00 (12.25–58.00)	23.00 (0.00–53.00)	0.110
	BP	461	46.00 (13.50–60.75)	35.00 (5.00–58.50)	0.073
Daily Hindrance (VAS)	SB	158	45.00 (19.00–52.00)	48.50 (19.75–58.00)	0.382
	DC	372	45.50 (20.00–63.00)	46.00 (15.50–60.00)	0.313
	CH	145	37.00 (12.50–50.00)	39.00 (14.00–51.50)	0.698
	FD	107	45.00 (25.00–53.00)	39.00 (14.25–53.75)	0.352
	**BP**	**459**	**50.00 (23.50–59.50)**	**38.50 (13.00–56.25)**	**0.007**

SB: sleep bruxism, DC: diurnal grinding or clenching, CH: chin on hand, FD: face down, BP: bad posture. In bold: Variables with significant difference.

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
