# Peer review of "Relationship between Oral Parafunctional and Postural Habits and the Symptoms of Temporomandibular Disorders: A Survey-Based Cross-Sectional Cohort Study Using Propensity Score Matching Analysis"

_jcm, 2022, doi:10.3390/jcm11216396_

Round 1

Reviewer 1 Report

Thank you for your comprehensive study that reviews a significant cohort of patients with temporo-mandibular joint disorders in an effort to elucidate potential causes and risk factors.

You have collected a significant amount of data and performed a robust analysis thereof. As you noted in your conclusion, further studies looking at occlusal status in conjunction with the other factors may prove to be a useful exercise in further identifying potential causes.

While no direct causative factors were identified, known or suspected risk factors were confirmed to play a role. 

In tables 2 and 3, you have a heading "Age" in the top left but it does not appear to relate to anything. Is this an error?

Author Response

Thank you for your comprehensive study that reviews a significant cohort of patients with temporomandibular joint disorders in an effort to elucidate potential causes and risk factors.

You have collected a significant amount of data and performed a robust analysis thereof. As you noted in your conclusion, further studies looking at occlusal status in conjunction with the other factors may prove to be a useful exercise in further identifying potential causes.

While no direct causative factors were identified, known or suspected risk factors were confirmed to play a role. 

(Response)

Thank you for your kind words.

In tables 2 and 3, you have a heading "Age" in the top left but it does not appear to relate to anything. Is this an error?

(Response)

We are sorry for your confusion. We would like to show the mean ages of the participants on the first line. Then, we revised “Age” to “Age (years)” and I added the number of patients in each result. (revisions: Tables 2, 3, and Supplementary Tables).

Reviewer 2 Report

This research and article is very well prepared and conducted. Habitual factors of musculosceletal disorders are hard to evaluate so the use of score matched patients pairs is sound method.

I has only one confusing moment: the questionnaire is reffered to be self-reporting. But in the disscusion section (line 358) is mentioned self awareness of sleep bruxism. Does it mean the possibility of reported bruxism recognition? 

Author Response

This research and article is very well prepared and conducted. Habitual factors of musculosceletal disorders are hard to evaluate so the use of score matched patients pairs is sound method.

(Response)

Thank you for your nice words.

I have only one confusing moment: the questionnaire is referred to be self-reporting. But in the discussion section (line 358) is mentioned self-awareness of sleep bruxism. Does it mean the possibility of reported bruxism recognition? 

(Response)

We are sorry for your confusion. We wanted to point out the difference between SB detected by the self-reporting questionnaire in our study and SB recognized by patient himself in previous study. This means that SB detected by the self-reporting questionnaire may include SB informed from others. (no revision)

Reviewer 3 Report

Dear Authors,

To investigate whether the presence of parafunctional habits (OPFHs) and postural habits (PHs) could predispose to Temporomandibular disorders (TMDs), authors performed a survey-based cross-sectional cohort study of self-reported TMDs among outpatients who visited the Temporomandibular Joint Clinic of Tokushima University Hospital. They aimed to assess the prevalence of self-reported subjective symptoms indicative of TMDs and their correlation with OPFHs and PHs associated with the onset and progression of TMDs.

The study was in line with the aims of the journal. 

However, there are some issues that should be addressed.

Abstract

The abstract section was well written. 

Introduction

I suggest improving the introduction section, which is too short and poor.

After definition, please report the classification according to Diagnostic Criteria for TMD (DC/TMD) Axis I. Thus, report that TMD could be divided in muscle disorders (including myofascial pain with and without mouth opening limitation) or intra-articular disorders (including disc displacement with or without reduction and mouth opening limitation, arthralgia, and arthritis). (cite and refer to: Schiffman E, Ohrbach R, Truelove E, et al. Diagnostic Criteria for Temporomandibular Disorders (DC/TMD) for Clinical and Research Applications: recommendations of the International RDC/TMD Consortium Network* and Orofacial Pain Special Interest Group. J Oral Facial Pain Headache. 2014;28(1):6-27.).

Moreover, improve epidemiological data, reporting that temporomandibular disorder is the second most common musculoskeletal disorder that causes pain and disability (cite and refer to: Valesan LF, Da-Cas CD, Réus JC, Denardin ACS, Garanhani RR, Bonotto D, Januzzi E, de Souza BDM. Prevalence of temporomandibular joint disorders: a systematic review and meta-analysis. Clin Oral Investig. 2021 Feb;25(2):441-453. doi: 10.1007/s00784-020-03710-w. Epub 2021 Jan 6. PMID: 33409693. And Jin LJ, Lamster IB, Greenspan JS, Pitts NB, Scully C, Warnakulasuriya S. Global burden of oral diseases: emerging concepts, management and interplay with systemic health. Oral Dis 2016; 22(7):609-19. Doi: 10.1111/odi.12428).

So, report the clinical manifestations as pain, decrease in the mouth opening, muscle or joint tenderness on palpation, limitation of mandibular movements, joint sounds and otologic complaints like tinnitus, vertigo or ear fullness, etcetera.

Then, report the commonly used treatments for both muscular and erthrogenous TMD (cite and refer to: Ferrillo M, Nucci L, Giudice A, Calafiore D, Marotta N, Minervini G, d'Apuzzo F, Ammendolia A, Perillo L, de Sire A. Efficacy of conservative approaches on pain relief in patients with temporomandibular joint disorders: a systematic review with network meta-analysis. Cranio. 2022 Sep 23:1-17. doi: 10.1080/08869634.2022.2126079. )

Among causes, report that central sensitization and sleep bruxism are main principal factors involved in TMD. (please cite and refer to: Ferrillo, M.; Giudice, A.; Marotta, N.; Fortunato, F.; Di Venere, D.; Ammendolia, A.; Fiore, P.; de Sire, A. PainManagement and Rehabilitation for Central Sensitization in Temporomandibular Disorders: A Comprehensive Review. Int. J. Mol. Sci. 2022, 23, 12164. doi: 0.3390/ijms232012164).

From line 48 is well written.

I suggest also to better define sleep bruxism, its diagnosis and its rule in TMD (Jiménez-Silva A, Peña-Durán C, Tobar-Reyes J, Frugone-Zambra R. Sleep and awake bruxism in adults and its relationship with temporomandibular disorders: A systematic review from 2003 to 2014. Acta Odontol Scand. 2017 Jan;75(1):36-58. doi: 10.1080/00016357.2016.1247465.)

Materials and Methods

Did you include patients with: evidence of concurrent illness; history of undergoing head-neck surgery; an ongoing orthodontic, speech-language or dental treatment or ongoing physical therapy treatment for neck pain; psychiatric diseases or neurological diseases?

Discussion and References were well written.

Author Response

To investigate whether the presence of parafunctional habits (OPFHs) and postural habits (PHs) could predispose to Temporomandibular disorders (TMDs), authors performed a survey-based cross-sectional cohort study of self-reported TMDs among outpatients who visited the Temporomandibular Joint Clinic of Tokushima University Hospital. They aimed to assess the prevalence of self-reported subjective symptoms indicative of TMDs and their correlation with OPFHs and PHs associated with the onset and progression of TMDs.

 The study was in line with the aims of the journal. 

However, there are some issues that should be addressed.

Abstract

The abstract section was well written. 

 (Response)

Thank you for your nice words.

Introduction

I suggest improving the introduction section, which is too short and poor.

After definition, please report the classification according to Diagnostic Criteria for TMD (DC/TMD) Axis I. Thus, report that TMD could be divided in muscle disorders (including myofascial pain with and without mouth opening limitation) or intra-articular disorders (including disc displacement with or without reduction and mouth opening limitation, arthralgia, and arthritis). (cite and refer to: Schiffman E, Ohrbach R, Truelove E, et al. Diagnostic Criteria for Temporomandibular Disorders (DC/TMD) for Clinical and Research Applications: recommendations of the International RDC/TMD Consortium Network* and Orofacial Pain Special Interest Group. J Oral Facial Pain Headache. 2014;28(1):6-27.).

Moreover, improve epidemiological data, reporting that temporomandibular disorder is the second most common musculoskeletal disorder that causes pain and disability (cite and refer to: Valesan LF, Da-Cas CD, Réus JC, Denardin ACS, Garanhani RR, Bonotto D, Januzzi E, de Souza BDM. Prevalence of temporomandibular joint disorders: a systematic review and meta-analysis. Clin Oral Investig. 2021 Feb;25(2):441-453. doi: 10.1007/s00784-020-03710-w. Epub 2021 Jan 6. PMID: 33409693. And Jin LJ, Lamster IB, Greenspan JS, Pitts NB, Scully C, Warnakulasuriya S. Global burden of oral diseases: emerging concepts, management and interplay with systemic health. Oral Dis 2016; 22(7):609-19. Doi: 10.1111/odi.12428).

So, report the clinical manifestations as pain, decrease in the mouth opening, muscle or joint tenderness on palpation, limitation of mandibular movements, joint sounds and otologic complaints like tinnitus, vertigo or ear fullness, etcetera.

Then, report the commonly used treatments for both muscular and erthrogenous TMD (cite and refer to: Ferrillo M, Nucci L, Giudice A, Calafiore D, Marotta N, Minervini G, d'Apuzzo F, Ammendolia A, Perillo L, de Sire A. Efficacy of conservative approaches on pain relief in patients with temporomandibular joint disorders: a systematic review with network meta-analysis. Cranio. 2022 Sep 23:1-17. doi: 10.1080/08869634.2022.2126079. )

Among causes, report that central sensitization and sleep bruxism are main principal factors involved in TMD. (please cite and refer to: Ferrillo, M.; Giudice, A.; Marotta, N.; Fortunato, F.; Di Venere, D.; Ammendolia, A.; Fiore, P.; de Sire, A. Pain Management and Rehabilitation for Central Sensitization in Temporomandibular Disorders: A Comprehensive Review. Int. J. Mol. Sci. 2022, 23, 12164. doi: 0.3390/ijms232012164).

From line 48 is well written.

I suggest also to better define sleep bruxism, its diagnosis and its role in TMD (Jiménez-Silva A, Peña-Durán C, Tobar-Reyes J, Frugone-Zambra R. Sleep and awake bruxism in adults and its relationship with temporomandibular disorders: A systematic review from 2003 to 2014. Acta Odontol Scand. 2017 Jan;75(1):36-58. doi: 10.1080/00016357.2016.1247465.)

 (Response)

Thank you for your special guidance. We have added several sentences you suggested in the Introduction section. Furthermore, we have cited several articles as references. (revisions: Lines 39-44, 50-56, and 59-61; References #5-7, 19, 20, and 25)

Materials and Methods

Did you include patients with: evidence of concurrent illness; history of undergoing head-neck surgery; an ongoing orthodontic, speech-language or dental treatment or ongoing physical therapy treatment for neck pain; psychiatric diseases or neurological diseases?

 (Response)

In this survey, the patients with systematic illnesses were excluded from the participants. (no revision)

Discussion and References were well written.

(Response)

Thank you for nice words.

Round 2

Reviewer 3 Report

Dear authors,

In my opinion the paper is ready for the publication.

I found this work impactful and fit well with in the scope of this journal.